# Bioactivities and Chemotaxonomy of Four *Heracleum* Species: A Comparative Study Across Plant Parts

**DOI:** 10.3390/ph18040576

**Published:** 2025-04-16

**Authors:** Tugce Ince Kose, Gamze Benli Yardimci, Damla Kirci, Derya Cicek Polat, Betul Demirci, Mujde Eryilmaz, Ceyda Sibel Kilic

**Affiliations:** 1Department of Pharmaceutical Botany, Graduate School of Health Sciences, Ankara University, 06560 Ankara, Türkiye; tikose@ankara.edu.tr; 2Department of Pharmaceutical Microbiology, Faculty of Pharmacy, Afyonkarahisar Health Sciences University, 03200 Afyonkarahisar, Türkiye; gamze.benli@afsu.edu.tr; 3Department of Pharmacognosy, Faculty of Pharmacy, Izmir Katip Çelebi University, 35620 Izmir, Türkiye; damla.kirci@ikc.edu.tr; 4Department of Pharmaceutical Botany, Faculty of Pharmacy, Ankara University, 06560 Ankara, Türkiye; polatd@ankara.edu.tr (D.C.P.); erdurak@pharmacy.ankara.edu.tr (C.S.K.); 5Department of Pharmacognosy, Faculty of Pharmacy, Anadolu University, 26210 Eskişehir, Türkiye; bdemirca@anadolu.edu.tr; 6Department of Pharmaceutical Microbiology, Faculty of Pharmacy, Acibadem Mehmet Ali Aydınlar University, 34638 Istanbul, Türkiye

**Keywords:** *Heracleum*, antibiofilm, antimicrobial, antioxidant, anti-quorum sensing, essential oils, phytochemical

## Abstract

**Background/Objectives**: This study investigates the phytochemical profile, essential oil composition, and bioactivities—including antioxidant, antimicrobial, antibio-film, and anti-quorum sensing (QS) activities—of four *Heracleum* L. species (*H. crenatifolium* Boiss, *H. paphlagonicum* Czeczott, *H. sphondylium* subsp. *montanum* Schleich. ex Gaudin, and *H. pastinacifolium* subsp. *incanum* (Boiss. & A.Huet) P.H.Davis). **Methods:** Total phenolic and flavonoid contents were quantified using the Folin–Ciocalteu and aluminum chloride colorimetric methods, respectively. Essential oils were extracted by hydrodistillation and analyzed via Gas Chromatography–Flame Ionization Detector (GC–FID) and Gas Chromatography–Mass Spectrometry (GC–MS), while Principal Component Analysis (PCA) and Hierarchical Cluster Analysis (HCA) evaluated chemical variability among the species. Antioxidant activities were assessed using DPPH and ABTS free radical scavenging assays. Antimicrobial activity was assessed using the broth microdilution method to determine Minimum Inhibitory Concentration (MIC) values, while antibiofilm activity was evaluated using an in vitro microplate-based biofilm model against *Pseudomonas aeruginosa* PAO1. Anti-QS activity was analyzed using a disc diffusion assay with *Chromobacterium violaceum* ATCC 12472 as the reporter strain. **Results:** It was observed that the amounts of total phenolic compounds and total flavonoids were higher in root extracts than in aerial parts extracts for the four species in this study (*H. sphondylium* subsp. *montanum* excluding phenolic content). In the analysis of essential oil, it was determined that the major component in the roots was mostly myristicin, and in the fruits it was mostly octyl acetate. Phenolic and flavonoid contents were positively correlated with antioxidant activity. Methanol and *n*-hexane extracts of *H. pastinacifolium* (aerial parts) and *n*-hexane extracts of *H. paphlagonicum* (root) exhibited notable antimicrobial activity, primarily against Gram-positive bacteria, but none of the extracts showed activity against *Klebsiella pneumoniae* ATCC 13383 or *P. aeruginosa* ATCC 27853. Among methanol extracts, *H. pastinacifolium* (aerial parts) exhibited the highest antibiofilm activity (73.2%), while *H. paphlagonicum* (aerial parts) showed the highest activity among *n*-hexane extracts (75.5%). All *n*-hexane extracts exhibited anti-QS activity, whereas the methanol extracts showed no activity. **Conclusions:** These findings underscore the chemical diversity and bioactive potential of *Heracleum* species, contributing to the chemotaxonomic understanding of the genus and supporting their potential applications in medicine and industry. To our knowledge, this is the first study that reveals the antibiofilm and anti-QS properties of these *Heracleum* species.

## 1. Introduction

The *Apiaceae* (previously known as the *Umbelliferae*) family, known as the Parsley-Celery family, is one of the largest families in the plant kingdom, comprising 300 to 455 genera and approximately 3000 to 3700 species worldwide [1,2,3]. Members of this family are characterized by their distinctive pungent or aromatic odor, attributed to the presence of essential oils or oleoresins [4]. Among *Apiaceae*, the genus *Heracleum* L. is notable for its diversity and widespread distribution, with approximately 125 taxa primarily found in the Asian continent. In Türkiye, 23 *Heracleum* taxa grow naturally, nine of which are endemics, giving the genus an impressive 41% endemism rate in the country [5,6,7,8].

The *Heracleum* genus is known for its traditional and modern uses [9]. In Türkiye, species of *Heracleum* are commonly referred to by various names, including Tavşancıl Otu, Kuruöğrek, Telehaş, Suh, Devesil, Öküz Havucu, and Herkül Otu [10,11]. These plants, often called “hogweed” in Europe, are among the tallest herbaceous plants, sometimes reaching heights of 4 to 5 m, and have been described with adjectives such as “large”, “tall”, or “giant” [12].

Species of *Heracleum* have been extensively used in traditional Asian medicine for their therapeutic properties. They have been used for a variety of different purposes such as antipyretic, analgesic, diaphoretic, antiseptic, carminative, and digestive, and also in rheumatic transmission, diarrhea, dysentery, lumbago, stomachache, fracture, fracture and fall injuries, as flavoring agents and spices. The fruits of this genus are also used as antiseptic, carminative, digestive, and analgesic in Iranian folk medicine [13,14]. Phytochemical investigations of *Heracleum* species have identified diverse secondary metabolites, including furanocoumarins, anthraquinones, stilbenes, flavonoids, hydrocarbons, monoterpenes, sesquiterpenes, coumarins, and steroids [15].

The essential oils of *Heracleum* species are complex mixtures of bioactive compounds, with significant variations in chemical composition depending on the species and plant parts utilized. These oils are predominantly characterized by the presence of terpenoids, phenylpropanoids, and various oxygenated compounds. Among the major constituents identified are octyl acetate, octyl hexanoate, octyl octanoate, β-pinene, myristicin, limonene, and hexyl butanoate, which collectively contribute to the unique aromatic and biological properties of the oils [16]. This diversity in chemical composition not only reflects the chemotaxonomic value of the *Heracleum* genus but also highlights its potential as a natural resource for bioactive compounds.

Reactive oxygen species (ROS), such as superoxide, peroxides, and hydroxyl radicals are known to cause oxidative stress, leading to conditions such as arthritis, atherosclerosis, and neurodegenerative diseases, including Alzheimer’s and Parkinson’s. Antioxidant compounds, often derived from medicinal plants, play a critical role in neutralizing these oxidizing agents. Secondary metabolites such as phenolic compounds, alkaloids, and tannins are particularly important in this regard [17].

The growing threat of antibiotic resistance has prompted urgent efforts to discover alternative antibacterial compounds. Resistant bacterial strains have become a global health crisis, causing significant morbidity and mortality [18,19]. Quorum sensing (QS), a bacterial communication system that regulates biofilm formation and virulence, has emerged as a promising target for addressing antibiotic resistance. Recent studies have highlighted the potential of anti-QS and antibiofilm molecules, particularly those derived from natural products, as alternatives to traditional antibiotics [20,21,22].

A review of the literature indicates that species in the genus *Heracleum* exhibit various biological activities. These include antioxidant, antimicrobial, antifungal, antiepileptic, analgesic, antipyretic, antihyperlipidemic, anxiolytic, anti-inflammatory, antiedema, cytotoxic, antidiabetic, anticonvulsant, and anti-Alzheimer activities. Several species within the *Heracleum* genus have demonstrated notable antimicrobial properties, attributed primarily to their essential oil compositions. For instance, the essential oil extracted from the flowering aerial parts of *H. moellendorffii* comprises 50 compounds, accounting for 92.67% of the oil’s composition, and exhibits significant antioxidant and antibacterial activities. Similarly, the essential oil of *Heracleum sphondylium* subsp. *ternatum*, rich in compounds such as 1-octanol (50.3%), octyl butyrate (24.6%), and octyl acetate (7.3%), has been evaluated for its antimicrobial activity using microdilution broth and agar diffusion methods, with 1-octanol identified as the bioactive constituent. These findings underscore the potential of *Heracleum* species as sources of bioactive compounds with antimicrobial efficacy. Additionally, *Heracleum* species have been utilized as odor enhancers and insecticides, showcasing their diverse potential in both medicinal and industrial contexts [13,23,24,25,26,27,28,29,30,31,32]. This study investigates the chemical compositions and biological activities of four *Heracleum* species growing in Türkiye, including the endemic *H. crenatifolium* Boiss, *H. paphlagonicum* Czeczott, and *H. sphondylium* subsp. *montanum* Schleich. ex Gaudin, as well as *H. pastinacifolium* subsp. *incanum* Boiss. & A.Huet species. This research aims to carry out the following: (i) Determine the total phenolic and flavonoid contents of methanol extracts; (ii) Analyze the essential oil composition of fruits, aerial parts, and roots of the selected *Heracleum* species; (iii) Evaluate the relationship between total phenolic compound and total flavonoid content with antioxidant activity of methanol extract; (iv) Investigate the antimicrobial, antibiofilm, and anti-QS activities of methanol and *n*-hexane extracts.

## 2. Results and Discussion

### 2.1. Determination of Total Phenolic and Total Flavonoid Content

The total phenolic and total flavonoid contents of eight methanol extracts (10 mg/mL), including the aerial parts and roots of *H. crenatifolium*, *H. paphlagonicum*, *H. sphondylium* subsp. *montanum*, and *H. pastinacifolium* subsp. *incanum* were determined (Appendix A). Additionally, the yield of each extract was calculated and given in Appendix A. See Appendix A.

The total phenolic content of methanol extracts prepared from the aerial parts and roots of *H. crenatifolium*, *H. paphlagonicum*, *H. sphondylium* subsp. *montanum*, and *H. pastinacifolium* subsp. *incanum* was determined using Folin–Ciocalteu reagent (FCR). Among the root extracts, *H. pastinacifolium* subsp. *incanum* exhibited the highest total phenolic content (335.744 mg GAEs/g extract). Similarly, among the aerial parts extracts, the highest phenolic content was observed in the *H. pastinacifolium* subsp. *incanum* (37.720 mg GAEs/g extract). In general, it was found that root extracts contained higher amounts of phenolic compounds than aerial parts extracts.

Total flavonoid amounts in methanol extracts prepared from the aerial parts and roots of *H. crenatifolium*, *H. paphlagonicum*, *H. sphondylium* subsp. *montanum*, and *H. pastinacifolium* subsp. *incanum* were determined using the aluminum chloride colorimetric method. Among the root extracts, the extract containing the highest total of flavonoids was *H. crenatifolium* (154.773 mg QE/g extract). Among the aerial parts extracts, the extract containing the highest amount of flavonoids was *H. pastinacifolium* subsp. *incanum* (70.696 mg QE/g extract). In general, it was observed that the amount of flavonoids was higher in root extracts than in aerial parts extracts.

In literature reviews, it was observed that there might be differences between the total phenol and total flavonoid amounts of the species, and it was thought that the reason for this change may be factors such as the collection locality of the plant, content differences in the subspecies of the plant, collection time, and the status of the plant within the vegetation cycle [33,34].

### 2.2. Essential Oils of Heracleum Genus

The chemical composition of essential oils extracted from three parts (fruits, aerial parts, and roots) of four *Heracleum* species (*H. crenatifolium*, *H. paphlagonicum*, *H. sphondylium* subsp. *montanum* and *H. pastinacifolium* subsp. *incanum*) were analyzed in this study. Across the analyzed plant parts, substantial variations in the main compounds and their yields were identified, emphasizing the nature of the essential oil profiles (Appendix A). See Appendix A. These findings contribute to the understanding of the phytochemical diversity within the genus *Heracleum*, known for its medicinal and aromatic properties.

Substantial interspecies and interparts variability were observed, particularly in terms of the major compounds and their yields. Additionally, the oil yields varied significantly, reflecting differences in essential oil productivity among the species and plant parts and also demonstrating the uneven distribution of secondary metabolites within and across the *Heracleum* species. The highest yield was recorded in the fruits of *H. sphondylium* subsp. *montanum* (4.39%), while the aerial parts of the same species showed a notably low yield (0.19%). Similarly, the roots of *H. crenatifolium* had a modest yield of 0.42%. This pattern of high oil productivity in fruits compared to other parts is consistent across the studied species, indicating that fruits are the primary sites for essential oil biosynthesis and accumulation. The essential oil profile of *H. crenatifolium* exhibited notable diversity across the fruits, aerial parts, and roots.

#### 2.2.1. Essential Oils Derived from Fruits

In *H. crenatifolium*, the main compound in the fruits’ essential oil was octyl acetate, which accounted for 95.4% of the total composition. The aerial parts were characterized by the presence of myristicin (10.6%), while the roots exhibited a distinctive profile, with myristicin being the major component (88.0%). Trace amounts of other monoterpenes such as *α*-pinene and *β*-pinene were also detected, highlighting the chemical compositions of this species.

In *H. paphlagonicum*, the fruits’ essential oil was found to contain octyl acetate (39.4%), with significant contributions from myristicin (15.1%) and hexyl butyrate (12.9%). The aerial parts showed a high presence of myristicin (42.9%), which was also the main compound in the roots’ essential oil, constituting 83.7% of the total composition. These results underscore the prevalence of myristicin in the non-fruit parts of this species, suggesting its potential as a key phytochemical marker.

The fruits’ essential oil of *H. sphondylium* subsp. *montanum* demonstrated a rich profile, dominated by octyl acetate (73.7%) and myristicin (8.9%). In contrast, the aerial parts exhibited a balanced distribution, with octyl acetate (56.1%) and hexyl butyrate (4.1%) as the main compounds. The roots’ essential oil was highly rich in myristicin (62.6%), further corroborating the trend of myristicin dominance in root oils across species. Additionally, the presence of limonene and octanol in minor amounts adds to the chemical diversity of this subspecies.

The chemical composition of *H. pastinacifolium* subsp. *incanum* showed significant variation across plant parts. The fruit oil was predominantly composed of hexyl butyrate (37.0%), followed by octyl acetate (15.6%). The aerial parts displayed a more complex profile, with myristicin (75.2%) as the major compound, alongside trace levels of methyl chavicol and octyl butyrate. In the root oil, myristicin accounted for 62.6%, consistent with the root-specific patterns observed in other species.

A comparative evaluation of the essential oils from all four species reveals significant interspecies and tissue-specific variations. Octyl acetate and myristicin emerged as the main compounds, with octyl acetate dominating fruit oils and myristicin being the main compound in roots. The fruits of *H. crenatifolium* showed the highest presence of octyl acetate, while *H. pastinacifolium* subsp. *incanum* displayed a more diverse profile with contributions from hexyl butyrate.

In aerial parts, myristicin consistently emerged as the main compound, reflecting a shared chemotype among the species. The roots showed a remarkable trend of myristicin enrichment, underscoring its potential functional significance. Despite these similarities, the presence of minor compounds such as limonene, hexanol, and methyl chavicol adds to the distinctive chemical fingerprints of each species. These results highlight the chemical compositions and adaptive strategies of the *Heracleum* genus.

A clear trend emerges from the comparative analysis of the essential oil yields and chemical compositions across the four *Heracleum* species (according to Appendix A). See Appendix A: (i) Fruits consistently yielded the highest amounts of essential oils, with octyl acetate as the main compound, particularly in *H. crenatifolium* and *H. sphondylium* subsp. *montanum*. (ii) Aerial parts showed the lowest yields across all species, with myristicin prevailing as the main compound. The limited diversity and low productivity may indicate a supplementary defensive role for the aerial parts. (iii) Roots demonstrated moderate yields with main consistent of myristicin in all species. This underscores the evolutionary importance of myristicin as a root-specific secondary metabolite, potentially serving as a chemical barrier against biotic stress. Despite these overarching similarities, the relative presences of minor compounds such as limonene, hexyl butyrate, and methyl chavicol varied among species and plants’ parts, contributing to their unique chemical signatures. These interspecies and inter-tissue differences underline the significant chemical plasticity within the *Heracleum* genus and its adaptive responses to ecological pressures.

The results of our study on the fruits of *Heracleum* species, along with the findings from the literature, are presented in Appendix A. See Appendix A. In our study, the essential oil obtained from the fruits of *H. crenatifolium* was determined to contain 95.4% octyl acetate. Similarly, the fruits of *H. sphondylium* subsp. *montanum* were found to contain 73.7% octyl acetate. In other fruit-focused studies in the literature, octyl acetate has been identified as the main component, with percentages generally ranging between 30 and 70%. For instance, essential oil from the fruits of *H. orphanidis* Boiss. in North Macedonia was reported to contain 84.5% octyl acetate, while in Iran, the octyl acetate content of *H. persicum* Desf. ex Fisch., C.A.Mey. & Avé-Lall fruits varied between 12.3% and 40.8% [35,36,37,38]. The high proportions of this compound in *Heracleum* fruits from Türkiye highlight their chemical richness and potential industrial applications.

Additionally, in our study, the fruits of *H. pastinacifolium* subsp. *incanum* were found to contain 37.0% hexyl butyrate and 20.6% octyl butanoate. This species shows notable diversity in its chemical profile compared to other *Heracleum* fruit studies reported in the literature. For example, essential oils from *H. persicum* fruits in Iran have been reported to contain hexyl butyrate in proportions ranging from 13.8% to 56.5%, alongside octyl acetate ranging between 12.3% and 40.8% [35,36,37]. The high levels of these components in our findings underscore the unique potential of *Heracleum* fruits from Türkiye, due to local environmental conditions and genetic factors [39].

In conclusion, the fruit-focused nature of our study provides a significant advantage when comparing it to other *Heracleum* essential oil studies in the literature. The high proportions of octyl acetate and hexyl butyrate found in our study suggest that the fruits are rich in essential oils, and these components derived from fruits could be valuable for biological and industrial applications. This underscores the importance of further focusing on the chemical profiles of essential oils from fruits.

#### 2.2.2. Essential Oils Derived from Aerial Parts

In our study, the essential oil compositions obtained from the aerial parts of *H. crenatifolium*, *H. paphlagonicum*, *H. sphondylium* subsp. *montanum*, and *H. pastinacifolium* subsp. *incanum*, collected from Türkiye, were analyzed. The results of these species are presented in the first four rows of Appendix A, and comparisons were made with data obtained from the literature. See Appendix A.

The aerial parts of *H. crenatifolium* were found to contain 85.9% octyl acetate and 10.6% myristicin. *H. sphondylium* subsp. *montanum* exhibited 56.1% octyl acetate and 24.2% myristicin. Similarly, the aerial parts of *H. pastinacifolium* subsp. *incanum* contained 62.6% myristicin, which is notably higher than the values reported for the aerial parts of *H. moellendorffii* Hance. [26]. Compared to these, *H. paphlagonicum* demonstrated a distinct chemical profile with 42.9% myristicin and 30.8% (*E*)-anethole.

According to the literature, *H. anisactis* Boiss. & Hohen from Iran has been reported to contain up to 93.5% myristicin, while *H. orphanidis* from Macedonia exhibited an octyl acetate content of 83.5%. The high octyl acetate levels observed in *H. crenatifolium* and *H. sphondylium* subsp. *montanum* suggest that this compound is produced in greater quantities under the environmental conditions of Türkiye. Additionally, the 30.8% (*E*)-anethole content in *H. paphlagonicum* exceeds the 25.0% reported for species such as *H. persicum* in the literature [40].

In conclusion, the essential oil compositions of *H. crenatifolium*, *H. paphlagonicum*, *H. sphondylium* subsp. *montanum*, and *H. pastinacifolium* subsp. *incanum* analyzed in this study stand out when compared with other *Heracleum* species in the literature, particularly due to their high levels of octyl acetate, myristicin, and (*E*)-anethole.

#### 2.2.3. Essential Oils Derived from Roots

In this study, the essential oil compositions obtained from the roots of *H. crenatifolium*, *H. paphlagonicum*, *H. sphondylium* subsp. *montanum*, and *H. pastinacifolium* subsp. *incanum*, collected from Türkiye, were analyzed. The results for these species are presented in the first four rows of Appendix A, while the remaining data were obtained from the literature. See Appendix A.

In our study, the root essential oil of *H. crenatifolium* was found to contain 88.0% myristicin. Similarly, the roots of *H. paphlagonicum* contained 83.7% myristicin, while *H. pastinacifolium* subsp. *incanum* exhibited notable richness with 75.2% myristicin. In the roots of *H. sphondylium* subsp. *montanum*, octyl acetate (57.1%) and myristicin (13.8%) were identified as the main compounds. These results indicate that myristicin is a common and main component in the roots across species, suggesting its potential role in the defensive mechanisms of the roots.

According to other studies in the literature, the root essential oils of *H. transcaucasicum* Manden. and *H. anisactis* from Iran contain 96.8% and 95.1% myristicin, respectively. In contrast, the roots of *H. orphanidis* from Macedonia were reported to contain 80.0% (*Z*)-falcarinol [41]. The high levels of myristicin observed in *H. crenatifolium* and *H. paphlagonicum* in our study highlight the influence of Türkiye’s ecological conditions in promoting the biosynthesis of this compound.

The literature data also reveal that certain species exhibit significant proportions of other compounds in their root essential oil profiles, such as *β*-pinene (17.7–35.1%), limonene (16.0–22.7%), and (*Z*)-falcarinol [38,41,42,43,44]. However, the major of myristicin in the species analyzed in our study underscores the chemical diversity of *Heracleum* roots and highlights the impact of genetic and environmental factors on root chemistry.

In conclusion, the root essential oil compositions of *H. crenatifolium*, *H. paphlagonicum*, *H. sphondylium* subsp. *montanum*, and *H. pastinacifolium* subsp. *incanum* analyzed in our study demonstrates a unique richness in myristicin compared to other *Heracleum* species reported in the literature. The predominance of this compound in roots suggests significant biological activity and potential relevance to plant defense mechanisms.

### 2.3. Multivariate Analysis of Heracleum Genus

PCA and HCA were used to characterize the volatile compounds of *Heracleum* spp. essential oils, with analyses conducted for the fruit, aerial parts, and root sections using Minitab 19 software [45]. The PCA numbers of the fruits’ essential oils for statistical analysis are given in Appendix A. See Appendix A. PCA analysis is illustrated in Figure 1, while HCA analysis is given in Figure 2. PCA was used to show the interrelationships among the species of the *Heracleum* fruits. In addition, the cluster obtained was confirmed by PCA analysis to evaluate the accuracy of this classification. For the essential oils of *Heracleum* fruits, all variables affected, PC1 (14.8%) and PC2 (13.7%), clarified 27.5% of the accumulated variation of the data analyzed.

The cluster analysis of the fruits’ essential oil observations (Figure 2) revealed two main clades, with similarity ranging from 40.56% to 99.99%. The essential oil of *H. crenatifolium* (PCA no: 1) used in this study was clustered with the essential oil of *H. crenatifolium* (Gümüşhane/Türkiye) (PCA no: 33), which suggested that these plants have a closely related origin (99.97%). The similarity between *H. paphlagonicum* (PCA no: 2) and *H. sphondylium* subsp. *montanum* (PCA no: 3) was determined to be 94.12%, while the similarity between *H. pastinacifolium* subsp. *incanum* (PCA no: 4) and PCA 3 was calculated as 92.84%. Among the four essential oils obtained in the study, it was found that the chemical compositions of *H. paphlagonicum* and *H. sphondylium* subsp. *montanum* were more like each other. Additionally, the essential oils were obtained using hydrodistillation (PCA no: 61) and microdistillation (PCA no: 62) methods [46]. The similarity between the oils obtained by these two methods was found to be 99.99%. This result indicates that the change in method does not have a significant effect on the chemical composition of the essential oils.

In conclusion, our study provides a detailed analysis of the essential oils of *H. crenatifolium*, *H. paphlagonicum*, *H. sphondylium* subsp. *montanum*, and *H. pastinacifolium* subsp. *incanum.* This analysis is presented alongside a comprehensive review of other *Heracleum* species, with a particular focus on evaluating their chemical variability using PCA and HCA. By incorporating both chemical and statistical data, our findings contribute valuable insights into the chemotaxonomic characteristics of the *Heracleum* genus. Additionally, this study complements previous research by providing a broader perspective on the chemical diversity within the genus.

The results indicate that while certain trends in the essential oil composition can be observed across the analyzed species, variations in specific chemical compounds highlight the complexity and variability inherent within the genus. These findings underscore the importance of using robust statistical tools, such as PCA and HCA, to explore relationships between species and identify chemotaxonomic markers. When viewed in conjunction with earlier studies, this integrated approach offers the potential to clarify phylogenetic relationships and taxonomic classifications within *Heracleum*.

Moreover, our research emphasizes the value of combining essential oil analyses with additional data sources, such as morphological, anatomical, and phytochemical studies. While the essential oil compositions presented in this study provide significant insights, they alone may not be sufficient to fully resolve taxonomic ambiguities within the genus. Future studies should aim to expand this work by examining other plant parts, such as fruits, and exploring secondary metabolites that are commonly found in the *Heracleum* genus. These broader analyses could help refine the understanding of interspecies relationships and support more precise taxonomic adjustments, where needed.

In summary, while this study is not conclusive in redefining the taxonomic hierarchy of the examined species, it provides a critical step toward better understanding the diversity and chemical complexity of the *Heracleum* genus. It also highlights areas where further research is required to fully elucidate the genus’s chemotaxonomic framework and phylogenetic relationships. This study thus represents a significant contribution to the growing body of knowledge on *Heracleum* species.

Appendix A shows the PCA numbers of the aerial parts’ essential oils used for statistical analysis. See Appendix A. Figure 3 presents the results of the PCA analysis, whereas Figure 4 depicts the outcomes of the HCA analysis. PCA was used to show the interrelationships among the species of the *Heracleum* aerial parts. In addition, the cluster obtained was confirmed by PCA analysis to evaluate the accuracy of this classification. For the essential oils of *Heracleum* aerial parts, all variables affected, PC1 (12.6%) and PC2 (10.4%), clarified 23.0% of the accumulated variation of the data analyzed.

The cluster analysis of the aerial parts’ essential oil observations (Figure 3) revealed two main clades, with similarity ranging from 34.44% to 99.99%. The essential oil of *H. crenatifolium* (PCA no: 1) used in this study was clustered with the essential oil of *H. orphanidis* (Macedonia) (PCA no: 20), which suggested that these plants have a closely related origin (99.70%) [41].

The similarity between *H. crenatifolium* (PCA no: 1) and *H. sphondylium* subsp. *montanum* (PCA no: 3) was determined to be 98.16%, while the similarity between *H. pastinacifolium* subsp. *incanum* (PCA no: 4) and *H. paphlagonicum* (PCA 2) was calculated as 95.89%. Among the four essential oils obtained in the study, it was found that the chemical compositions of *H. crenatifolium* and *H. sphondylium* subsp. *montanum* were more like each other. The lowest similarity, calculated as 34.44%, was observed between *H. paphlagonicum* from Türkiye (PCA no: 1) and *H. sprengelianum* from India (PCA no: 22). This notable difference can be attributed to several factors, including both geographic and species-related variations. Geographic location plays a critical role in shaping the chemical composition of essential oils due to differences in environmental conditions such as climate, soil type, and altitude. Furthermore, the distinct taxonomic characteristics of the two species contribute significantly to the observed variation. These findings highlight the combined influence of ecological and genetic factors on the chemical profiles of essential oils derived from different *Heracleum* species across diverse regions.

Appendix A shows the PCA numbers of the roots’ essential oils used for statistical analysis. See Appendix A. Figure 5 presents the results of the PCA analysis, while Figure 6 displays the outcomes of the HCA analyses. PCA was used to show the interrelationships among the species of the *Heracleum* roots. In addition, the cluster obtained was confirmed by PCA analysis to evaluate the accuracy of this classification. For the essential oils of *Heracleum* roots, all variables affected, PC1 (18.8%) and PC2 (17.2%), clarified 36.0% of the accumulated variation of the data analyzed.

The cluster analysis of the roots’ essential oil observations (Figure 5) revealed two main clades, with similarity ranging from 35.19% to 99.99%. The essential oil of *H. crenatifolium* (PCA no: 1) used in this study was clustered with the essential oil of *H. paphlagonicum* (PCA no: 2), which suggested that these plants have a closely related origin (99.61%). Also, the essential oil of PCA no: 2 used in this study was clustered with the essential oil of *H. pastinacifolium* subsp. *incanum* (PCA no: 4), which suggested that these plants have a closely related origin (99.94%).

The similarity rate between *H. crenatifolium* PCA no: 1 and *H. stevenii* (PCA no: 11) was calculated to be 35.19% based on the analyses conducted. This finding indicates a limited shared variance between the analyzed components, thereby suggesting a significant difference in the chemical profiles of the two components. Specifically, PCA no: 11 was identified as belonging to the species *H. stevenii*, which was derived from the essential oil of this species grown in Russia [42]. This information supports broader analyses of regional and species–specific variations in the chemical composition of the essential oil in question, reaffirming the utility of PCA as an effective tool for evaluating such chemical profiles. Furthermore, the similarity rate of 35.19% highlights the potential impact of diverse environmental or genetic factors on the chemical profile.

### 2.4. Antioxidant Activity

For DPPH free radical scavenging activity, all extracts (8 methanol extracts) were prepared separately at 7 different concentrations (1000–10 μg/mL). IC_50_ values of all, including gallic acid used as standard, were calculated and given in Table 1.

For ABTS free radical scavenging activity, all extracts (8 methanol extracts) were prepared at a concentration of 1 mg/mL, and antioxidant activities of all extracts were calculated as mg Trolox equivalent/g extract and given in Table 1.

No antioxidant activity studies have been found on *H. paphlagonicum* and *H. crenatifolium* species, and thus this study was the first to evaluate the aforementioned bioactivity. The DPPH free radical scavenging activity of *H. crenatifolium*, *H. paphlagonicum*, *H. sphondylium* subsp. *montanum*, and *H. pastinacifolium* subsp. *incanum* aerial parts and root methanol extracts was evaluated according to their DPPH bleaching abilities. The most active extract was *H. paphlagonicum* root extract among root extracts; the most active extract among aerial parts extracts was *H. crenatifolium* aerial parts extract.

*H. crenatifolium*, *H. paphlagonicum*, *H. sphondylium* subsp. *montanum*, and *H. pastinacifolium* subsp. *incanum*, aerial parts and root methanol extracts were evaluated according to the ABTS radical cation scavenging method. The highest trolox equivalent ABTS scavenging effect was in *H. sphondylium* subsp. *montanum* root extract among root extracts; the highest effect was in *H. paphlagonicum* aerial parts extract among aerial parts extracts.

In general, when the DPPH free radical scavenging activity and ABTS free radical scavenging activity results were evaluated, the root extracts of *Heracleum* species showed higher antioxidant activity than aerial parts extracts. It was observed that having a high extract yield had no effect in terms of antioxidant activity. In our study and literature reviews, it was seen that the total phenol and total flavonoid contents of the species were directly proportional to the antioxidant activity, and it was thought that the different antioxidant activities of the species in our study in other literature reviews were due to differences in total phenol and total flavonoid contents [33,34,47].

### 2.5. Antimicrobial Activity

The MIC values (mg/mL) of the methanol and *n*-hexane extracts of four *Heracleum* species are shown in Table 2. The MIC values of four *Heracleum* species ranged from 0.625 to 10 mg/mL against the tested bacteria and 5 to 10 mg/mL against the tested fungus. Among the tested extracts, the methanol and *n*-hexane extracts of *H. pastinacifolium* subsp. *incanum* (aerial parts) and *n*-hexane extract of *H. paphlagonicum* (root) exhibited the best antimicrobial activity. The extracts demonstrated better antibacterial activity against Gram-positive bacteria. However, none of the extracts showed antibacterial activity against *Klebsiella pneumoniae* ATCC 13383 and *Pseudomonas aeruginosa* ATCC 27853. The observed activities can be considered weak compared to standard antimicrobials (Table 2).

The phytochemical analysis of methanol extracts from *Heracleum* species demonstrated significant variability in total phenolic and flavonoid contents, with root extracts generally exhibiting higher levels than aerial parts.

Among the aerial parts extracts, *H. pastinacifolium* subsp. *incanum* exhibited the highest phenolic content (37.720 mg GAEs/g extract) and was also among the most active extracts against Gram-positive bacteria, including *S. aureus* and MRSA. Phenolic compounds are well documented for their antimicrobial properties, primarily through mechanisms such as bacterial cell wall disruption, inhibition of essential enzymes, and interference with nucleic acid synthesis. The strong activity of *H. pastinacifolium* subsp. *incanum* aerial parts extracts against Gram-positive pathogens may be attributed to their high phenolic content, as these bacteria lack the outer membrane barrier present in Gram-negative bacteria, making them more susceptible to phenolic compounds [48,49,50].

Similarly, the aerial parts extracts of *H. pastinacifolium* subsp. *incanum*, which exhibited the highest flavonoid content (70.696 mg QE/g extract) among the aerial parts extracts, showed notable antimicrobial activity. Flavonoids have been reported to exert their antimicrobial effects by disrupting bacterial membranes, and inhibiting efflux pumps [51].

The weak or absent activity of the extracts against Gram-negative bacteria aligns with the structural and functional barriers posed by the outer membrane of these bacteria [52]. While phenolic and flavonoid compounds are effective against Gram-positive pathogens, their efficacy against Gram-negative bacteria often requires higher concentrations or synergistic combinations with other bioactive compounds [53].

When comparing the species included in this study with the existing literature, their antimicrobial activities were found to align with previously reported findings [28,29,54]. The phytochemical properties of *Heracleum* species, particularly their high phenolic and flavonoid contents, underpins their antimicrobial properties. The correlation between phytochemical composition and antimicrobial activity reinforces the therapeutic potential of these plants. Future studies should focus on isolating and characterizing individual bioactive compounds to better understand their mechanisms of action and to explore their synergistic potential with existing antimicrobials.

### 2.6. Antibiofilm Activity

The percentage biofilm inhibition values of the methanol extracts of four *Heracleum* species (10 mg/mL) are presented in Figure 7, while the percentage biofilm inhibition values of the *n*-hexane extracts of four *Heracleum* species (10 mg/mL) are presented in Figure 8. Among the methanol extracts, *H. pastinacifolium* subsp. *incanum* (aerial parts) exhibited the highest antibiofilm activity, whereas *H. paphlagonicum* (aerial parts) showed the highest activity among *n*-hexane extracts, with biofilm inhibition values of 73.2% and 75.5%, respectively. Appendix A present the optical density (OD) values and standard deviations (SD) corresponding to the biofilm inhibition percentages displayed in Figure 7 and Figure 8, respectively. See Appendix A.

The strong antibiofilm activity observed in the methanol extracts of *H. pastinacifolium* subsp. *incanum* (aerial parts) aligns with its high phenolic and flavonoid content. Phenolic compounds are known to interfere with bacterial adhesion and QS pathways, both of which are critical for biofilm formation [55]. Flavonoids, similarly, can disrupt bacterial communication and inhibit enzymes necessary for biofilm matrix development [50,56]. These findings suggest that the phenolic and flavonoid constituents in *H. pastinacifolium* subsp. *incanum* methanol extracts play a significant role in their antibiofilm activity.

Biofilm-associated infections pose a substantial challenge in medical settings, particularly in relation to implanted medical devices, chronic wounds, and respiratory tract infections [57]. The strong antibiofilm activity of *H. pastinacifolium* subsp. *incanum* (aerial parts) methanol extract positions them as potential candidates for further development into antibiofilm agents. To the best of our knowledge, this is the first study investigating the antibiofilm activity of the four *Heracleum* species included in this research (*H. crenatifolium*, *H. paphlagonicum*, *H. sphondylium* subsp. *montanum*, and *H. pastinacifolium* subsp. *incanum*). This study highlights the distinctive bioactive properties of these species, particularly their ability to inhibit biofilm formation—a feature that has not been previously investigated.

### 2.7. Anti-Quorum Sensing Activity

The anti-QS activity test was carried out based on the MIC values of the extracts of four *Heracleum* species against *C. violaceum* ATCC 12472 (Appendix A). See Appendix A. No violacein inhibition zone was observed in methanol extracts, but it was observed in all *n*-hexane extracts. Based on the anti-QS activity results, *n*-hexane extracts exhibited better activity.

## 3. Materials and Methods

### 3.1. Plant Material

The *Heracleum* species used in this study—*H. crenatifolium*, *H. paphlagonicum*, *H. sphondylium* subsp. *montanum*, and *H. pastinacifolium* subsp. *incanum*—were collected from the localities given in Table 3 (fruiting period). The plant materials were identified by Tugçe Ince Kose, Prof. Dr. Hayri Duman, and Prof. Dr. Ahmet Duran. Voucher specimens were deposited at the herbarium of Ankara University Faculty of Pharmacy (AEF).

### 3.2. Preparation of Extracts

#### 3.2.1. Methanol Extract

Plant extracts were obtained using the maceration method. A total of 100 g of plant material (roots and aerial parts) was ground using a grinder (Renas Makine ve Otomasyon Teknolojileri RBT-200, İstanbul, Türkiye). The ground material was macerated three times for 8 h with methanol at room temperature using magnetic stirrers (Heidolph MR 3001, Darmstadt, Germany), (VELP ARE, VELP AREC.X, Usmate Velate, Italy). The extracts were filtered using Filtros Anonia, S.A. filter paper (Barcelona, Spain), and the methanol was evaporated under reduced pressure at a maximum temperature of 65 °C using a rotary evaporator (Büchi Rotavapor R-200, Flawil, Switzerland).

#### 3.2.2. *n*-Hexane Extract

*n*-Hexane extracts were prepared using a Soxhlet apparatus. Approximately 25–30 g of plant material (roots and aerial parts) was placed in the Soxhlet apparatus cartridge, and *n*-hexane (Emplura^®^, Darmstadt, Germany) was added and flushed once. The extraction process was carried out for 4 h, after which the solvent was removed using a rotary evaporator. Due to insufficient plant material, *n*-hexane extracts could not be prepared from the roots of *H. pastinacifolium* subsp. *incanum* and *H. crenatifolium*.

### 3.3. Phytochemical Composition

#### 3.3.1. Determination of Total Phenolic Compounds

The total phenolic content of the extracts was determined using the FCR method, following the protocol outlined by Büyüktuncel [58]. Gallic acid, at concentrations ranging from 50 to 500 μg/mL, was used as a standard to construct the calibration curve. Absorbance measurements were taken at 765 nm using a spectrophotometer (Shimadzu UV Spectrophotometer UV-1800, Kyoto, Japan). The total phenolic content was calculated as gallic acid equivalents (GAE) based on the calibration curve and expressed as milligrams of GAE per gram of extract (mg GAE/g extract). Each experiment was performed in triplicate to ensure reproducibility and reliability.

#### 3.3.2. Determination of Total Flavonoids

The total flavonoid content of the extracts was determined using the aluminum chloride colorimetric method, as described by Demir et al. [59] and Keskin et al. [60]. Quercetin, at concentrations ranging from 50 to 800 μg/mL, was used as a standard to construct the calibration curve. Absorbance was measured at 510 nm using a spectrophotometer. Based on the quercetin calibration curve, the total flavonoid content was calculated and expressed as milligrams of quercetin equivalents (mg QE) per gram of extract (mg QE/g extract). All experiments were performed in triplicate to ensure accuracy and reproducibility.

### 3.4. Essential Oils

Essential oils were isolated from the fruits, roots, and aerial parts of the *Heracleum* species and analyzed for their chemical compositions using Gas Chromatography with Flame Ionization Detection (GC–FID) and Gas Chromatography–Mass Spectrometry (GC–MS). To assess the variation in essential oil compositions across different *Heracleum* species, Hierarchical Cluster Analysis (HCA) and Principal Component Analysis (PCA) were performed.

#### 3.4.1. Isolation of the Essential Oils

Air-dried fruits, aerial parts, and roots of the *Heracleum* species (fruiting period) were subjected to hydrodistillation for 3 h using a Clevenger-type apparatus, following the protocol recommended by the [61]. Obtained essential oils were dried over anhydrous sodium sulfate to remove moisture and stored in sealed vials at +4 °C, protected from light, to preserve their quality until further analysis and testing.

#### 3.4.2. GC–MS Analysis

GC–MS analysis was carried out with an Agilent 5975 GC-MSD system (Santa Clara, CA, USA). Innowax FSC column (60 m × 0.25 mm, 0.25 mm film thickness) was used with helium as carrier gas (0.8 mL/min). The GC oven temperature was initially maintained at 60 °C for 10 min, then increased to 220 °C at a rate of 4 °C/min, where it was held constant for 10 min. Subsequently, the temperature was increased to 240 °C at a rate of 1 °C/min. The split ratio was set to 40:1, and the injector temperature was adjusted to 250 °C. Mass spectra were recorded at 70 eV with a mass range of m/z 35 to 450 [62].

#### 3.4.3. GC–FID Analysis

GC analysis was carried out using an Agilent 6890N GC system. The flame ionization detector (FID) temperature was set to 300 °C. To obtain the same elution order on the GC–MS, simultaneous auto-injection was carried out on the same type of column (Innowax FSC column, 60 m × 0.25 mm, 0.25 μm film thickness) under identical operational conditions. The relative percentage amounts of the separated compounds were calculated from FID chromatograms, respectively [62].

#### 3.4.4. Identification of the Volatile Compounds

The identification of essential oil components was performed by comparing their relative retention times with those of authentic reference samples or by comparing their relative retention indices (RRIs) to a series of n-alkanes. Computer matching against commercial (Wiley GC/MS Library, MassFinder Software 4.0), and in-house “Başer Library of Essential Oil Constituents” built up by genuine compounds and components of known oils [62].

#### 3.4.5. Statistical Analysis

Statistical analysis was performed to evaluate the chemical variability among the essential oils. Original variables were defined as essential oil contents that exceeded 3.0% of the overall essential oil composition in at least one species. After normalization, the data was subjected to HCA and PCA. Minitab 19 was used to conduct the statistical analysis (State College, PA, USA). Using the rescaled distances in the dendrogram and a cut-off point (Euclidean distance) that allows the attainment of consistent clusters, the number of clusters was computed. To determine the similarity among the essential oils regarding the contents of their chemical constituents, the PCA and the HCA were utilized [45].

### 3.5. Antioxidant Activity

#### 3.5.1. DPPH Free Radical Scavenging Activity

The DPPH free radical scavenging effect of the extracts was evaluated according to their DPPH bleaching ability. The experiment is based on the DPPH redox change reaction [63]. The reaction mixture contained 100 μM of DPPH dissolved in methanol and extracts at different concentrations. The absorbance values of the test substances kept at room temperature for 30 min were measured at 517 nm and the free radical capture percentages were calculated. The experiment was performed in triplicate. Gallic acid was used as a reference. Calibration curves were determined for gallic acid and each sample according to the % inhibition formula given below. IC50 values were determined according to these curves for gallic acid and all samples.

The free radical scavenging effect was calculated according to the % inhibition formula:% Inhibition = [(Acontrol* − Asample*/Acontrol*)] × 100

Acontrol*: Absorbance of control; Asample*: Absorbance of extract

#### 3.5.2. ABTS Free Radical Scavenging Activity

Total antioxidant activity was determined using the standard TEAC (Trolox equivalent antioxidant capacity) method [64]. ABTS used during the experiment is a mixture of 2.45 mM potassium persulfate and 7 mM ABTS aqueous solution. The mixture was left to stand in the dark at room temperature overnight (12 to 16 h) and blue-green colored radical formation was achieved. The obtained ABTS radical cation was diluted with methanol (pH 7.4). The samples were diluted with ABTS solution at a ratio of 1/10. After waiting for 6 min, the absorbance value at 734 nm was measured and the amount of inhibition was calculated.

Trolox (water-soluble α-tocopherol analogue) was used as a standard. Calibration curve was obtained according to 5 different concentrations and absorption values of Trolox. Antioxidant activity of 8 different extracts was calculated according to this calibration curve. The results were given as Trolox equivalent antioxidant capacity (TEAC). The experiment was performed in triplicate.

### 3.6. Antimicrobial Activity

The antimicrobial activity of the extracts was assessed using the broth microdilution method to determine the MIC values, following the guidelines provided by the Clinical and Laboratory Standards Institute [65,66]. The test organisms included *S. aureus* ATCC 25923, methicillin-resistant *S. aureus* ATCC 43300 (MRSA), *E. coli* ATCC 25922, *P. aeruginosa* ATCC 27853, *K. pneumoniae* ATCC 13883 as bacterial strains, and *C. albicans* ATCC 10231 as the fungal strain. The plant extracts were dissolved in 5% dimethyl sulfoxide (DMSO), and all experiments were conducted in triplicate to ensure reproducibility.

For the antibacterial activity test, serial two-fold dilutions of the extracts, ranging from 10 to 0.78 mg/mL, were prepared in Mueller Hinton Broth (Merck, Darmstadt, Germany). Inoculums were prepared from 24-h subcultures, and the final bacterial suspension concentration was adjusted to 5 × 10^5^ CFU/mL. After incubation at 35 °C for 18–24 h, the MIC value (µg/mL) was determined as the lowest concentration that completely inhibited visible bacterial growth. Negative controls consisted of inoculated broth with 5% DMSO, while ciprofloxacin and gentamicin were used as standard antibiotics.

For the antifungal activity test, serial two-fold dilutions of the extracts (10 to 0.78 mg/mL) were prepared in RPMI 1640 broth (ICN-Flow, Aurora, OH, USA). The final fungal suspension concentration was adjusted to 0.5 to 2.5 × 10^3^ CFU/mL. The microplate was incubated at 35 °C for 48 h. The MIC value (µg/mL) was defined as the lowest concentration that completely inhibited visible fungal growth. Negative controls consisted of 5% DMSO and RPMI 1640 broth, with fluconazole serving as the standard antifungal agent.

### 3.7. Antibiofilm Activity

The antibiofilm activity of the extracts against *Pseudomonas aeruginosa* PAO1 was evaluated using the crystal violet assay, based on an in vitro microplate-based biofilm model. Before the test, the MIC values of the extracts were determined against *P. aeruginosa* PAO1 [20,21].

For biofilm formation, *P. aeruginosa* PAO1 was cultured in Brain Heart Infusion (BHI) broth at 37 °C for 24 h. Final inoculum suspensions (∼10^6^ CFU/mL) were prepared in BHI enriched with 2% sucrose. In total, 100 µL of the inoculum suspensions were added to 96-well microtiter plates for all test conditions. The plates were incubated at 37 °C for 72 h to allow the formation of mature biofilms.

After biofilm formation, the medium was aspirated, and non-adherent cells were removed by washing the wells with sterile phosphate-buffered saline (PBS, pH 7.2). Subsequently, 100 µL of plant extracts were added to each well containing the mature *P. aeruginosa* biofilms. The plates were incubated at 37 °C for 24 h. After incubation, the wells were washed with PBS to remove non-adherent cells and dried at room temperature for 1 h. An amount of 100 µL of 0.5% crystal violet solution was added to each well to stain the biofilm cells. After 30 min, the wells were washed three times with PBS, and a 30:70 (*v/v*) acetone-alcohol solution was added to dissolve the bound dye within the biofilm matrix. BHI broth enriched with 2% sucrose was used as the control. The optical density (OD) of the dissolved crystal violet was measured at 620 nm using a microplate reader (Thermo Scientific Multiskan GO Microplate Spectrophotometer, Vantaa, Finland). The percentage biofilm inhibition values were calculated using the following formula:Biofilm Inhibition (%) = [(OD (Growth control) − OD (Sample))/OD (Growth control)] × 100

Data were represented as means ± SD. Statistical differences were analyzed using One-way ANOVA and Tukey’s test for post-hoc comparisons with GraphPad Prism software (version 8.0, GraphPad, Boston, MA, USA).

### 3.8. Anti-Quorum Sensing Activity

The anti-quorum sensing (anti-QS) activity of the plant extracts was evaluated using the disc diffusion method with *Chromobacterium violaceum* ATCC 12472 as the reporter strain, following the protocols described by Çiçek Polat et al. [21] and Junejo et al. [22]. Before the test, the MIC values of the extracts against *C. violaceum* ATCC 12472 were determined.

An overnight bacterial culture was prepared, and its density was adjusted to 1.5 × 10^8^ CFU/mL. The bacterial suspension was then inoculated onto Luria-Bertani (LB) agar plates. Sterile blank discs (6 mm diameter; Bioanalyse^®^, Ankara, Turkey) were impregnated with 20 µL of the plant extracts and placed on the agar surface. The plates were incubated at 30 °C for 24 h. Following the incubation at 30 °C for 24 h, the plates were examined for the presence of a violacein inhibition zone. The formation of an inhibition zone around the discs indicated positive anti-QS activity.

## 4. Conclusions

This study presents a comprehensive investigation into the bioactivities and chemotaxonomic characteristics of four *Heracleum* species (*H. crenatifolium, H. paphlagonicum*, *H. sphondylium* subsp. *montanum*, and *H. pastinacifolium* subsp. *incanum*), three of which are endemic to our country, focusing on their phytochemical profiles, essential oil compositions, and some bioactivities across different plant parts.

The root extracts consistently demonstrated higher phenolic and flavonoid contents, correlating with their antioxidant activities. Among the essential oils, while it was demonstrated that octyl acetate dominated the essential oil of the fruits, myristicin was prevalent in the roots, highlighting potential ecological and functional adaptations and defining the distinctive chemical fingerprints of these species. The study also demonstrates the utility of PCA and HCA in elucidating chemotaxonomic relationships and interspecies chemical variability, offering valuable insights for further research in phylogenetics and taxonomy of the *Heracleum* genus.

The observed antimicrobial, antibiofilm, and anti-QS activities further underscore the pharmaceutical potential of these species, particularly against Gram-positive bacteria and biofilm-forming pathogens.

Future studies should expand on these findings by exploring additional plant parts, such as seeds, and investigating seasonal and geographic variations in chemical composition. Integrating morphological, anatomical, and genomic data is crucial for understanding taxonomic relationships within the genus and maximizing the pharmaceutical and industrial potential of *Heracleum* species.

## Figures and Tables

**Figure 1 pharmaceuticals-18-00576-f001:**
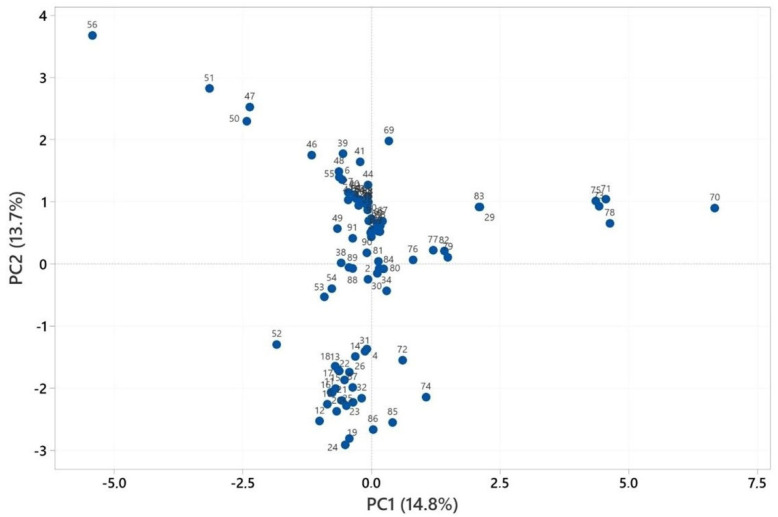
PCA analysis of the major components of the essential oils of fruits.

**Figure 2 pharmaceuticals-18-00576-f002:**
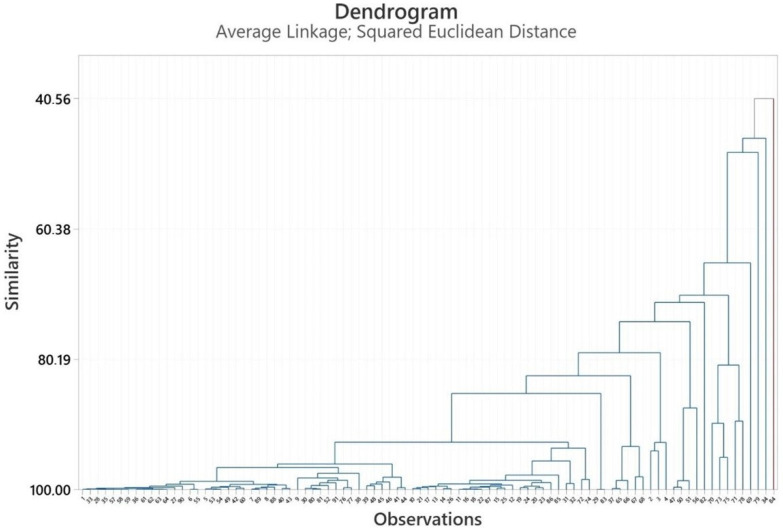
Dendrogram obtained by HCA based on Euclidian distances between groups of the major components of the fruits’ essential oils.

**Figure 3 pharmaceuticals-18-00576-f003:**
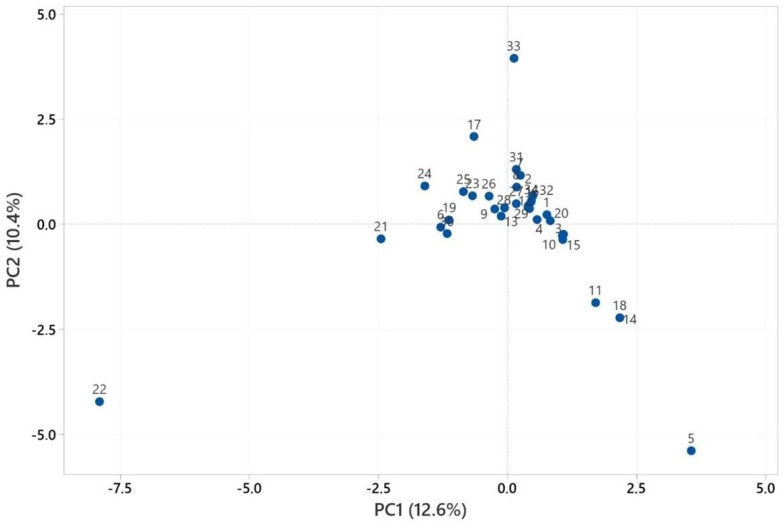
PCA analysis of the major components of the essential oils of aerial parts.

**Figure 4 pharmaceuticals-18-00576-f004:**
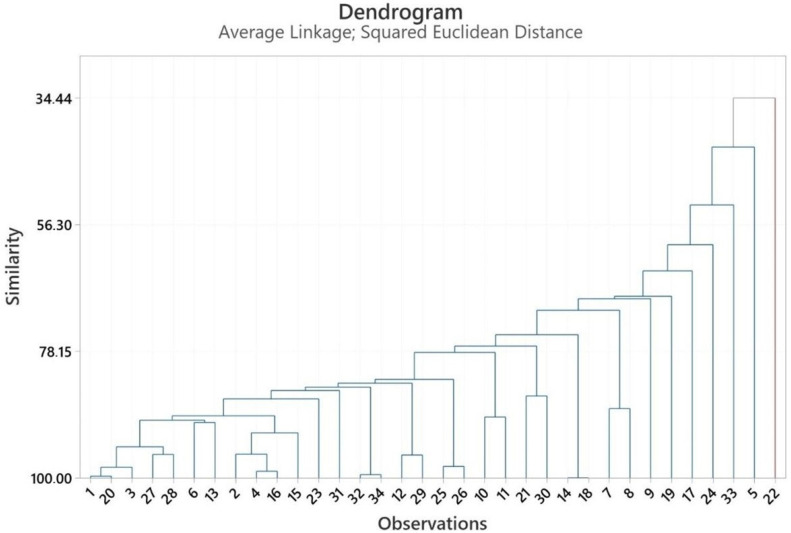
Dendrogram obtained by HCA based on Euclidian distances between groups of the major components of the essential oils of aerial parts.

**Figure 5 pharmaceuticals-18-00576-f005:**
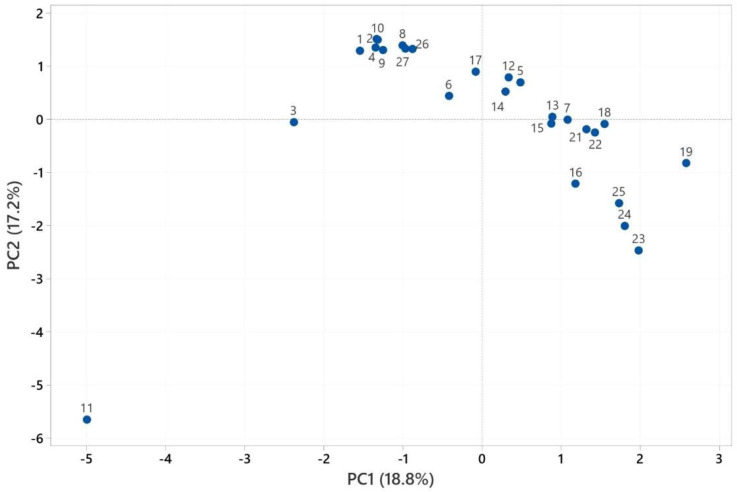
PCA analysis of the major components of the essential oils of roots.

**Figure 6 pharmaceuticals-18-00576-f006:**
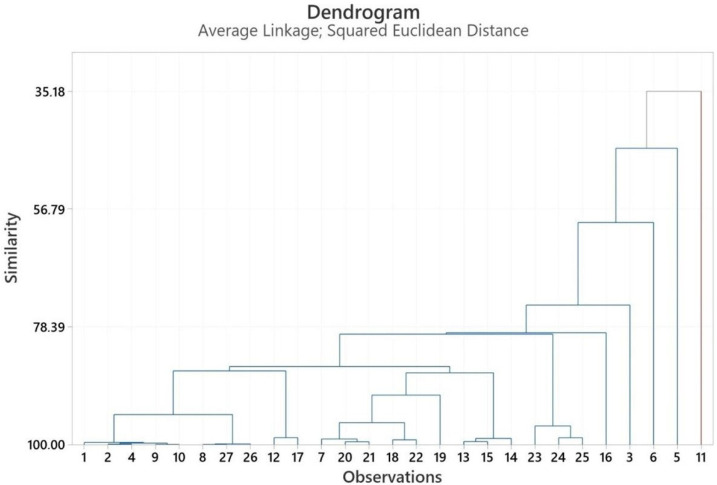
Dendrogram obtained by HCA based on Euclidian distances between groups of the major components of the roots’ essential oils.

**Figure 7 pharmaceuticals-18-00576-f007:**
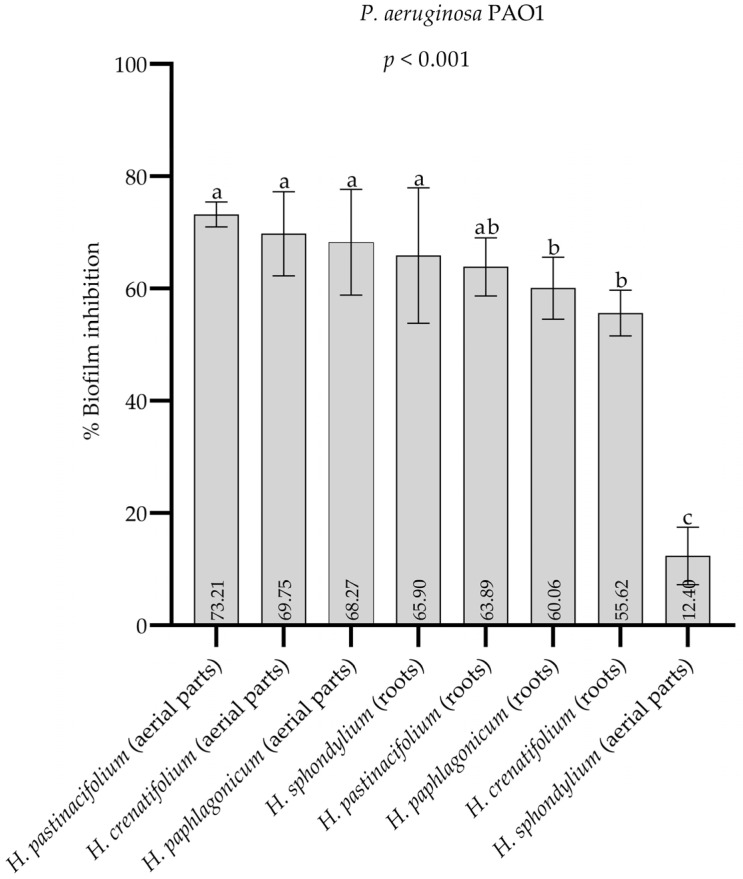
The percentage biofilm inhibition values of the methanol extracts of four *Heracleum* spe-cies. Data were represented as mean values with standard deviations. Different letters indicate the statistical difference. One-way ANOVA and Tukey’s tests were conducted to compare mean values of % biofilm inhibition between groups.

**Figure 8 pharmaceuticals-18-00576-f008:**
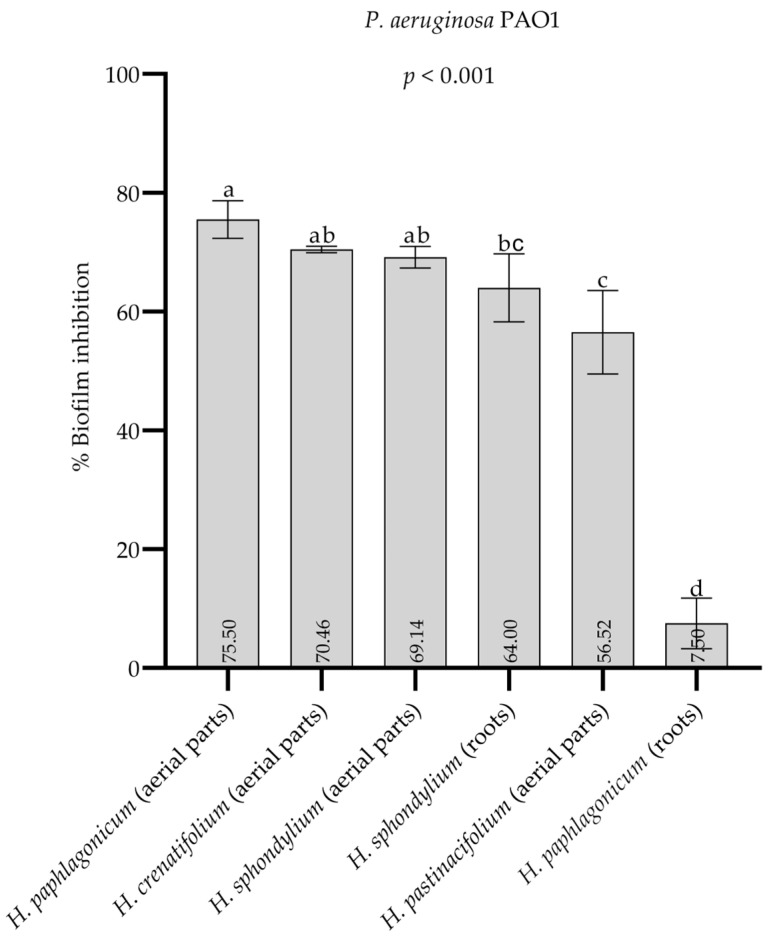
The percentage biofilm inhibition values of the *n*-hexane extracts of four *Heracleum* species. Data were represented as mean values with standard deviations. Different letters indicate the statistical difference. One-way ANOVA and Tukey’s tests were conducted to compare mean values of % biofilm inhibition between groups.

**Table 1 pharmaceuticals-18-00576-t001:** Antioxidant activity of methanol extracts.

Plant Extracts		DPPH Radical Scavenging Activity (IC_50_ Value (mg/mL) ± SD ^b^)	ABTS Free Radical Scavenging Activity (TEAC ^a^ mg TE/g Extract ± SD ^b^)
*H. crenatifolium*	aerial parts	0.216 ± 0.0007	155.622 ± 3.47
	roots	0.765 ± 0.015	143.574 ± 3.47
*H. paphlagonicum*	aerial parts	0.275 ± 0.001	157.63 ± 3.47
	roots	0.242 ± 0.004	199.799 ± 6.95
*H. sphondylium* subsp. *montanum*	aerial parts	0.246 ± 0.002	115.461 ± 6.95
	roots	0.416 ± 0.004	282.128 ± 6.95
*H. pastinacifolium* subsp. *incanum*	aerial parts	1.077 ± 0.007	79.317 ± 3.47
	roots	0.77 ± 0.003	135.542 ± 6.024
Gallic acid		0.001 ± 0.0001	-

^a^ TEAC: Trolox equivalent antioxidant capacity, ^b^ SD: Standard deviation.

**Table 2 pharmaceuticals-18-00576-t002:** MIC values (mg/mL) of the methanol and *n*-hexane extracts of four *Heracleum* species.

Plant Extracts		Gram-Positive Bacteria		Gram-Negative Bacteria			Fungus
		*S.* *aureus*	*S.* *aureus* (MRSA)	*E.* *coli*	*K. pneumoniae*	*P. aeruginosa*	*C. albicans*
Aerial parts (Methanol)	*H. crenatifolium*	1.25	1.25	-	-	-	-
	*H. paphlagonicum*	1.25	1.25	-	-	-	-
	*H. sphondylium* subsp. *montanum*	1.25	1.25	-	-	-	-
	*H. pastinacifolium* subsp. *incanum*	1.25	0.625	5	-	-	5
Roots (Methanol)	*H. crenatifolium*	2.5	2.5	-	-	-	10
	*H. paphlagonicum*	1.25	1.25	-	-	-	10
	*H. sphondylium* subsp. *montanum*	1.25	1.25	10	-	-	5
	*H. pastinacifolium* subsp. *incanum*	2.5	2.5	-	-	-	5
Aerial parts (*n*-Hexane)	*H. crenatifolium*	-	-	-	-	-	-
	*H. paphlagonicum*	1.25	1.25	-	-	-	-
	*H. sphondylium* subsp. *montanum*	1.25	1.25	-	-	-	-
	*H. pastinacifolium* subsp. *incanum*	0.625	0.625	-	-	-	10
Roots (*n*-Hexane)	*H. paphlagonicum*	0.625	0.625	-	-	-	-
	*H. sphondylium* subsp. *montanum*	1.25	5	-	-	-	-
Ciprofloxacin		<0.00025	0.0005	<0.00025	<0.00025	<0.00025	NT
Gentamicin		0.0005	<0.00025	0.0005	<0.00025	0.0005	NT
Fluconazole		NT	NT	NT	NT	NT	0.00156
5% DMSO		-	-	-	-	-	-

NT: not tested.

**Table 3 pharmaceuticals-18-00576-t003:** Localities and herbarium registry numbers of collected *Heracleum* species.

Plant	Locality	Date of Collection	Herbarium Number
*H. crenatifolium*	Konya; Hadim 810–820 m 37.0282, 32.6936	15.07.2023	AEF 31001
*H. paphlagonicum*	Kastamonu; Ilgaz Dağı 1800–1850 m 41.0772, 33.7299	22.07.2023	AEF 30999
*H. sphondylium* subsp. *montanum*	Ankara; Çamlıdere 1100–1150 m 40.5436, 32.5636	23.07.2023	AEF 31000
*H. pastinacifolium* subsp. *incanum*	Karabük; Keltepe 1950–2000 m 41.0866, 32.4623	05.08.2023	AEF 30998

## Data Availability

The data that support the findings of this study are available from the corresponding author upon reasonable request. The data are not publicly available due to ethical reasons, as they are part of an ongoing related project and currently under restricted access.

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
