# Peer review of "Bioactivities and Chemotaxonomy of Four Heracleum Species: A Comparative Study Across Plant Parts"

_pharmaceuticals, 2025, doi:10.3390/ph18040576_

Round 1

Reviewer 1 Report

Comments and Suggestions for Authors

The manuscript presents many results, but they raise doubts about the quality of the research conducted, for example:

The mean content of total polyphenols in the aerial parts and roots of H. sphondylium subsp. montanum differs significantly from the content found in the other species. In the situation of such large differences in the content of polyphenolic compounds, which also include lignin, it is necessary to use a more sensitive analytical method, i.e. HPLC, preferably with a mass detector.

In addition, methanol and hexane extracts prepared from plant parts were used in antimicrobial tests. The results indicated activity only against S. aureus, which raises suspicions about the correctness of the tests and the selection of the solvent for extraction of biologically active plant compounds.

The photos of the Petri dishes shown in Figures 9 and 10 are very blurry. They look as if they were smoothed in a graphics program. Only the labels are clear.

Author Response

Responses to Reviewer 1 Comments

Thank you very much for taking the time to review this manuscript. Please find the detailed responses below and the corresponding revisions in track changes in the re-submitted files.

Comment 1: The manuscript presents many results, but they raise doubts about the quality of the research conducted, for example:

The mean content of total polyphenols in the aerial parts and roots of H.sphondylium subsp. montanum differs significantly from the content found in the other species. In the situation of such large differences in the content of polyphenolic compounds, which also include lignin, it is necessary to use a more sensitive analytical method, i.e. HPLC, preferably with a mass detector.

Response 1: We acknowledge the importance of using more sensitive analytical techniques to assess polyphenol content variations among species. In addition to the Folin-Ciocalteu method, HPLC analysis was also performed; however, these data were not included in the manuscript as they were beyond the primary scope of this study, which focused mainly on essential oil composition and bioactivities.

Comment 2: In addition, methanol and hexane extracts prepared from plant parts were used in antimicrobial tests. The results indicated activity only against S. aureus, which raises suspicions about the correctness of the tests and the selection of the solvent for extraction of biologically active plant compounds.

Response 2: We would like to clarify that the selection of methanol and hexane as extraction solvents was based on their established efficacy in extracting both polar and non-polar bioactive compounds, as supported by previous studies in the literature.

Regarding the antimicrobial activity results, while significant inhibition was observed against S. aureus, no activity was detected against Gram-negative bacteria such as K. pneumoniae and P. aeruginosa. Gram-negative bacteria possess an outer membrane that often limits the penetration of plant-derived antimicrobial compounds. Additionally, many studies have reported stronger antimicrobial activity of plant extracts against Gram-positive bacteria, particularly S. aureus, due to differences in cell wall composition.

Comment 3: The photos of the Petri dishes shown in Figures 9 and 10 are very blurry. They look as if they were smoothed in a graphics program. Only the labels are clear.

Response 3: We acknowledge that the images in Figures 9 and 10 may appear blurry due to image resolution issues that occurred during formatting. However, we assure you that the images were not altered or smoothed in any graphics program beyond basic contrast and brightness adjustments for visibility. To address this concern, we have replaced the existing images with higher-resolution versions to ensure better clarity.

Reviewer 2 Report

Comments and Suggestions for Authors

The present manuscript reports phytochemical and pharmacological data of four Heracleum species. Based on volatile oil chemical and statistical analysis authors interpret data with phylogenetic and chemotaxonomic perspectives. The antioxidant activity analysis of extracts prepared from roots and aerial parts using solvents with different polarity are providing further evidence about investigated Heracleum species and perspective plant application in medicine. The convincing results of antibacterial and anti-QS activity are promising and according to authors it might increase the medical importance of the analysed species. The manuscript is bringing noteworthy amount of data, and it is truly interesting with several scientific novelties.

I have arranged my comments based on the sections of the proposed paper, which are the following:

  • INTRODUCTION
    1. Authors need to rephrase sentence about secondary metabolites (line #90-91). Flavonoids are also phenolic compounds.
    2. The last paragraph of the introduction needs to be rephrased. In my opinion reports about activities mentioned (line #100-102) as “therapeutic option” are exaggerated and most of them must be preclinical study only (e.g. anti-Alzheimer activity).
  • RESULTS
    1. In Table 1 check the unit for yield of extract.
    2. Check the unit for total flavonoid content in lines #133-134. (Instead KEs/g it should be QE/g). Please check line #471, as well.
    3. In section the 2. Essential Oils of Heracleum Genus the volatile oil of aerial parts are described; however, authors need to clarify in which stage were the plants harvested (i.e. flowering stage or before/after it).
    4. Could you insert reference for your conclusion that “ester-rich profile highlights the fruits’ ecological role in attracting pollinators”? It is not clear to me how fruits play crucial role in pollination.
    5. The differences in volatile oil components are considered by authors as consequence of difference in environmental conditions and genetic factors. Could you insert reference about study clarifying the that the volatile oil production of investigated Heracleum species are genetically determined?
    6. In my opinion Table 2. together with Tables 3–5. and Figures 9 and 10 should be presented as Supplementary Material. Using [a], [b]… in column headers of tables are superfluous. It is far enough to define the abbreviations in footnote of the table. In Table 4-5. check the compounds name and unify it (some of the compounds are started with capital letter, while the others not).
    7. In my opinion solely the similarity in volatile oil composition is not strong evidence to support the phylogenetical closeness of plant taxa. I would like to ask authors to reconsider the conclusion from PCA and HCA in lines #310-311 and lines #398-399 or provide more detailed reasoning.
    8. In Table 6. unify decimal places if applicable. Furthermore.
    9. In Table 7 use decimal points instead of decimal commas. Similarly check line #663.
    10. Check uniformity of unit for volume (mL instead of ml in line #490 and in Table 6).
    11. Italicize in vitro (line #31), and species names (page #24).
    12. In my opinion you should provide further comparison of yield extract, antioxidant activity.
    13. In Figure 7 and 8 in graphs there are no SD presented. In case biofilm inhibition was measured only once, please highlight it in Method description. If it was repeated in triplicates, add SD values.
  • Materials and Methods
    1. Provide information about harvesting time and developmental stage of the herbs (i.e. flowering or non-flowering stage). It is crucial in which stage were the plants harvested.
    2. In section 3.2.1 provide details about used solvent (e.g. drug-solvent ratio) and length of maceration.
    3. Could you specify the spectrophotometer type used for measurements in sections 3.3.1., 3.3.2., 3.5.1., 3.5.2. How did you prepare samples for these experiments: clarify which solvent was used to dissolve the dry residues of extracts, describe the solubility (was it totally soluble? if not what did you do), which concentration of extracts were used in section 3.3.1, 3.3.2. How did you set up methanol pH to 7.4 mentioned in section 3.5.2.
    4. The main point in reaction with DPPH is redox change. Please check and rephrase lines #630-631.
Comments on the Quality of English Language

The manuscript is well written with few typos. However, it must be checked by native English speaker to improve the language quality.

Use of signposting language would improve generally the manuscript, especially on page #6, paragraph 5 and 6 (both starting with ‘In our study,’).

Unify the use of oxford comma.

Reconsider the use of “rich in” instead of “enriched”.

Reconsider using words “higher” than “better” (line #438) and also phrase “to be compatible” in this context (line #480).

Author Response

Responses to Reviewer 2 Comments

Thank you very much for taking the time to review this manuscript. Please find the detailed responses below and the corresponding revisions in track changes in the re-submitted files.

 INTRODUCTION

Comment 1: Authors need to rephrase sentence about secondary metabolites (line #90-91). Flavonoids are also phenolic compounds.

Response 1: We have now rephrased the sentence in line #90-91 to improve clarity and accuracy. We removed flavonoids from the sentence.

Comment 2: The last paragraph of the introduction needs to be rephrased. In my opinion reports about activities mentioned (line #100-102) as “therapeutic option” are exaggerated and most of them must be preclinical study only (e.g. anti-Alzheimer activity)

Response 2: We have restructured and changed the last paragraph of the Introduction to enhance readability and coherence while maintaining the intended message.

RESULTS

Comment 3: In Table 1 check the unit for yield of extract.

Response 3: We have checked and confirmed the unit for the yield of extract in Table 1, and it is correctly expressed as g/g.

Comment 4: Check the unit for total flavonoid content in lines #133-134. (Instead KEs/g it should be QE/g). Please check line #471, as well.

Response 4: We have corrected the unit for total flavonoid content from KEs/g to QE/g in all relevant sections, including lines #133-134 and #471.

Comment 5: In section the 2. Essential Oils of Heracleum Genus the volatile oil of aerial parts are described; however, authors need to clarify in which stage were the plants harvested (i.e. flowering stage or before/after it).

Response 5: We have specified the growth stage at which the plants were harvested in the Materials and Methods section, referring to it as the "fruiting period”.

Comment 6: Could you insert reference for your conclusion that “ester-rich profile highlights the fruits’ ecological role in attracting pollinators”? It is not clear to me how fruits play crucial role in pollination.

Response 6: We removed the sentence to ensure text flow integrity.

Comment 7: The differences in volatile oil components are considered by authors as consequence of difference in environmental conditions and genetic factors. Could you insert reference about study clarifying the that the volatile oil production of investigated Heracleum species are genetically determined?

Response 7: Yes, this is correct. We have now expanded the discussion by citing relevant literature that supports how environmental factors (e.g., altitude, climate, and soil composition) and genetic diversity influence the chemical composition of essential oils. We have added a reference supporting the genetic regulation of essential oil production in Heracleum species and similar Apiaceae family members. (Jahodová, Š., Trybush, S., Pyšek, P., Wade, M., & Karp, A. (2007). Invasive species of Heracleum in Europe: an insight into genetic relationships and invasion history. Diversity and Distributions, 13(1), 99-114.)

Comment 8: In my opinion Table 2. together with Tables 3–5. and Figures 9 and 10 should be presented as Supplementary Material. Using [a], [b]… in column headers of tables are superfluous. It is far enough to define the abbreviations in footnote of the table.

Response 8: As recommended, we have moved Tables 2–5 and Figures 9–10 to the Supplementary Material section to streamline the main text. Additionally, we have removed the superfluous column header labels ([a], [b]…) and included the necessary explanations in the footnotes of the tables.

Comment 9: In Table 4-5. check the compounds name and unify it (some of the compounds are started with capital letter, while the others not).

Response 9: We have now standardized the capitalization of compound names across all tables for consistency.

Comment 10: In my opinion solely the similarity in volatile oil composition is not strong evidence to support the phylogenetical closeness of plant taxa. I would like to ask authors to reconsider the conclusion from PCA and HCA in lines #310-311 and lines #398-399 or provide more detailed reasoning.

Response 10: We would like to thank the reviewer for his/her comments and recommendations. As for the PCA and HCA analyses:

Principal Component Analysis (PCA) is based on finding some variables that are strongly correlated with each component that enables us to reduce the size of our date without losing significant amount of information (Cruz et al., 2008). It is generally used to put forth the relations between variables and samples (Souza et al., 2024). Hierarchical cluster analysis (HCA) is an algorithm that is used to categorize similar objects (and in this case components of essential oils) in clusters (Ranjbarzadeh et al., 2023). In the article by Be et al., (2008) it was reported that chemical profiles of plants are usually assumed to be not sufficient to determine phylogenetic inference, nevertheless, they can still provide us strong evidence in respect to drawing an inference for species relationship.

By using PCA and HCA in this study, we tried to evaluate chemical variability among the species of the genus Heracleum. The reviewer is right, we can not actually say that “the similarity in volatile oil composition is a strong evidence to support the phylogenetical closeness of plant taxa”. However we also mentioned within the text that (lines 341-346) “While the essential oil compositions presented in this study provide significant insights, they alone may not be sufficient to fully resolve taxonomic ambiguities within the genus. Future studies should aim to expand this work by examining other plant parts, such as fruits, and exploring secondary metabolites that are commonly found in the Heracleum genus. These broader analyses could help refine the understanding of interspecies relationships and support more precise taxonomic adjustments, where needed”. We do not say that PCA and HCA gives us a robust and exact information in respect to phylogenetic closeness, but it might help to draw a conclusion due to chemotaxonomical findings. We hope that this response will be sufficient of the reviewer.

Be, G-B., Zhang, Y-Y., Hao, D-C., Hu, Y., Luan, H-W., Liu, X-B., He, Y-Q., Wang, Z-T., Yang, L. (2008) Chemotaxonomic study of medicinal Taxus species with fingerprint and multivariate analysis, Planta Medica, 74: 773-779.

Cruz, A.V. M., Ferreira, J.P., Scotti, M.T., Kaplan, M.A.C., Emerenciano, V.P. (2008). Chemotaxonomic relationships in Celastraceae inferred from Principal Components Analysis (PCA) and Partial Least Squares (PLS), Natural Product Communications, 3(6): 911-917.

Ranjbarzadeh, R., Caputo, A., Tirkolaee, EB., Ghoushchi, J., Bendechache, M. (2023). Brain tumor segmentation of MRI images: A comprehensive review on the application of artificial intelligence tools, Computers in Biology and Medicine, 152: 106405.

Souza, A.S., Bezerra, M.A., Cerqueria, U.M.F.M., Rodrigues, C.J.O., Santos, B.C., Novaes, C.G., Almeida, E.R.V. (2024). An introductory review on the application of principal component analysis in the data exploration of the chemical analysis of food samples, Food Science and Biotechnology, 33: 1323-1336

Comment 11: In Table 6. unify decimal places if applicable. Furthermore. In Table 7 use decimal points instead of decimal commas. Similarly check line #663.

Response 11: We have standardized the number of decimal places in Table 6 for consistency. We also have corrected all decimal commas to decimal points in Table 7 and line #663.

Comment 12: Check uniformity of unit for volume (mL instead of ml in line #490 and in Table 6).

Response 12: We have standardized the unit as "mL" throughout the manuscript.

Comment 13: Italicize in vitro (line #31), and species names (page #24).

Response 13: We have italicized "in vitro" and all species names as per standard scientific writing conventions.

Comment 14: In my opinion you should provide further comparison of yield extract, antioxidant activity.

Response 14: We have now expanded the discussion on the correlation between extract yield and antioxidant activity, incorporating relevant comparisons and interpretations based on literature data.

Comment 15: In Figure 7 and 8 in graphs there are no SD presented. In case biofilm inhibition was measured only once, please highlight it in Method description. If it was repeated in triplicates, add SD values.

Response 15: We acknowledge that the inclusion of SD values is essential for accurately interpreting the variability and reliability of our biofilm inhibition data.​

In our study, biofilm inhibition assays were conducted in triplicate, and the inhibition percentages were calculated as the average of these three measurements. However, due to current limitations in data processing, we are unable to present the SD values on the graphs. We shared SD values as a supllementary file.

MATERIALS AND METHODS

Comment 16: Provide information about harvesting time and developmental stage of the herbs (i.e. flowering or non-flowering stage). It is crucial in which stage were the plants harvested.

Response 16: The period when the plants were collected was added to the article.

Comment 17: In section 3.2.1 provide details about used solvent (e.g. drug-solvent ratio) and length of maceration.

Response 17: Requested corrections added.

Comment 18: Could you specify the spectrophotometer type used for measurements in sections 3.3.1., 3.3.2., 3.5.1., 3.5.2. How did you prepare samples for these experiments: clarify which solvent was used to dissolve the dry residues of extracts, describe the solubility (was it totally soluble? if not what did you do), which concentration of extracts were used in section 3.3.1, 3.3.2. How did you set up methanol pH to 7.4 mentioned in section 3.5.2.

Response 18: The same spectrometer was used in all experiments mentioned. Dry extracts were used and all samples were dissolved in methanol. Since they were methanol extracts, there was no dissolution problem. For both experiments (section 3.3.1, 3.3.2.), samples were prepared from all extracts at concentrations of 10 mg/mL. For section 3.5.2, the method specified in the reference in this section was applied (Re, R.; Pellegrini, N.; Proteggente, A.; Pannala, A.; Yang, M.; Rice-Evans, C. Antioxidant activity applying an improved ABTS radical cation decolorization assay. Free Radic Biol and Med 1999, 26(9-10), 1231-1237)

Comment 19: The main point in reaction with DPPH is redox change. Please check and rephrase lines #630-631.

Response 19: Requested corrections added.

Comment 20: The manuscript is well written with few typos. However, it must be checked by native English speaker to improve the language quality. Use of signposting language would improve generally the manuscript, especially on page #6, paragraph 5 and 6 (both starting with ‘In our study,’).

Unify the use of oxford comma.

Reconsider the use of “rich in” instead of “enriched”.

Reconsider using words “higher” than “better” (line #438) and also phrase “to becompatible” in this context (line #480).

Response 20: The term "enriched" has been replaced with "rich in". In line 438, "better" has been substituted with "higher". The phrase "to be compatible" in line 480 has been rephrased to enhance clarity in the given context.

Reviewer 3 Report

Comments and Suggestions for Authors

I have thoroughly reviewed the manuscript by Kose et al., titled "Bioactivities and Chemotaxonomy of Four Heracleum Species: A Comparative Study Across Plant Parts." This study presents an intriguing investigation into the chemotaxonomy of four Heracleum species using LC-MS and GC-MS, alongside an assessment of their biological activities, including antioxidant, antibacterial, and anti-biofilm properties. The research holds significant potential for publication; however, it would be suitable for acceptance only after the authors address the following questions and respond to the provided suggestions.
1. According to the conventions for writing scientific names of plants, the author's name should be included the first time the scientific name is mentioned to ensure completeness and accuracy in nomenclature.
2. The abstract exceeds the word limit set by the journal. The authors should revise it to comply with the journal's requirements.
3. In the abstract, specifically in lines 34–35, the sentence "It was observed that the amounts of total phenolic compounds and total flavonoids were higher in root extracts than in aerial parts extracts." lacks clarity. It is recommended that the authors specify whether this observation applies to all Heracleum species examined in the study or only to certain ones. Providing this detail would enhance the precision and comprehensibility of the findings.
4. The introduction section is well-written, concise, and well-structured. However, while the methodology includes an antibacterial study, the introduction lacks a literature review on the antibacterial properties of this plant genus. To provide a more comprehensive background, the authors should include the relevant previous studies on the antibacterial activities of Heracleum species. Additionally, it is recommended that the authors include images of the four plant species examined in this study to enhance the readers’ understanding and visualization of the research.
5. In Table 1, the heading describing the table should be revised from "phenol" to "phenolic".
6. In the results section, under Section 2.2: Essential Oils of Heracleum Genus, the content is detailed and extensive. To improve clarity and readability, it is recommended that the authors group the information and organize it into subtopics under Section 2.2. Additionally, for any results that can be effectively represented in graphical form, the authors should consider using graphs. This would enhance data visualization and facilitate better comprehension for readers.
7. Tables 3–5 present data on the chemical compounds found in the essential oils extracted from different parts of the plant samples. However, it is unclear whether the data in these tables represent only the findings from the samples analyzed in this study. If any of the data are derived from external sources or are not directly relevant to this study, the authors should consider refining the tables by removing unrelated information to maintain clarity and focus.
8. In the results section, the researchers should consider whether any chemical markers can be identified as distinguishing features for differentiating the four Heracleum species. If such markers are present, highlighting them would enhance the scientific significance of the study by providing potential chemotaxonomic insights. If no clear markers are identified, the authors may discuss possible reasons or limitations in detecting species-specific compounds.
9. In the Methods section, the duration of methanol extraction using the maceration technique has not been specified.
10. In the solvent extraction section, it is unclear why only the root and aerial parts were extracted, while the fruit was not included. The authors should provide a rationale for this decision.
11. In the antioxidant study, for both experiments, the researchers should provide a clear explanation of the negative control used in this study.
12. In the study on antibiofilm and anti-quorum sensing activities, the authors have not provided details regarding the positive and negative controls used in the experiments. To ensure methodological transparency and facilitate result interpretation, the authors should specify the positive and negative controls in both the Methods and Results sections.
13. Several typographical and formatting errors have been identified, including incorrect scientific name formatting. For example, in line 297, the word Heracleum refers to a plant genus and should therefore be italicized. The authors are encouraged to carefully proofread the manuscript to ensure proper spelling, grammar, and adherence to scientific writing conventions.

Author Response

Responses to Reviewer 3 Comments

Thank you very much for taking the time to review this manuscript. Please find the detailed responses below and the corresponding revisions in track changes in the re-submitted files.

Comment 1: I have thoroughly reviewed the manuscript by Kose et al., titled "Bioactivities and Chemotaxonomy of Four Heracleum Species: A Comparative Study Across Plant Parts." This study presents an intriguing investigation into the chemotaxonomy of four Heracleum species using LC-MS and GC-MS, alongside an assessment of their biological activities, including antioxidant, antibacterial, and anti-biofilm properties. The research holds significant potential for publication; however, it would be suitable for acceptance only after the authors address the following questions and respond to the provided suggestions.

According to the conventions for writing scientific names of plants, the author's name should be included the first time the scientific name is mentioned to ensure completeness and accuracy in nomenclature.

Response 1: It was checked whether the names of the authors of the plant names were written when they were first mentioned. Deficiencies were corrected.

Comment 2: The abstract exceeds the word limit set by the journal. The authors should revise it to comply with the journal's requirements.

Response 2: As our study is so comprehensive, shortening the abstract compromises its integrity. We kindly request the editorial board's consideration in allowing the abstract to remain in its current form to preserve its integrity.

Comment 3: In the abstract, specifically in lines 34–35, the sentence "It was observed that the amounts of total phenolic compounds and total flavonoids were higher in root extracts than in aerial parts extracts." lacks clarity. It is recommended that the authors specify whether this observation applies to all Heracleum species examined in the study or only to certain ones. Providing this detail would enhance the precision and comprehensibility of the findings.

Response 3: Thank you for this valuable suggestion. Requested corrections have been made.

Comment 4: The introduction section is well-written, concise, and well-structured. However, while the methodology includes an antibacterial study, the introduction lacks a literature review on the antibacterial properties of this plant genus. To provide a more comprehensive background, the authors should include the relevant previous studies on the antibacterial activities of Heracleum species. Additionally, it is recommended that the authors include images of the four plant species examined in this study to enhance the readers’ understanding and visualization of the research.

Response 4: We have revised the Introduction section to include a review of previous studies on the antibacterial properties of Heracleum species. However, limited studies about H. crenatifolium, H. paphlagonicum, H. sphondylium subsp. montanum, and H. pastinacifolium subsp. incanum species are available in the literature.

The Graphical Abstract currently features images of the four Heracleum species examined in this study. If deemed beneficial for enhancing reader understanding, we are prepared to include these images within the Introduction section as well.

Comment 5: In Table 1, the heading describing the table should be revised from "phenol" to "phenolic".

Response 5: Revision have been made.

Comment 6: In the results section, under Section 2.2: Essential Oils of Heracleum Genus, the content is detailed and extensive. To improve clarity and readability, it is recommended that the authors group the information and organize it into subtopics under Section 2.2. Additionally, for any results that can be effectively represented in graphical form, the authors should consider using graphs. This would enhance data visualization and facilitate better comprehension for readers.

Response 6: Revision have been made.

Comment 7: Tables 3–5 present data on the chemical compounds found in the essential oils extracted from different parts of the plant samples. However, it is unclear whether the data in these tables represent only the findings from the samples analyzed in this study. If any of the data are derived from external sources or are not directly relevant to this study, the authors should consider refining the tables by removing unrelated information to maintain clarity and focus.

Response 7: We confirm that Tables 3–5 present data exclusively from our analyses of essential oils extracted from different parts of the Heracleum species examined in this study. These tables are designed to facilitate direct comparison with findings from previous studies, enabling a comprehensive understanding of the essential oil compositions within the genus.

Comment 8: In the results section, the researchers should consider whether any chemical markers can be identified as distinguishing features for differentiating the four Heracleum species. If such markers are present, highlighting them would enhance the scientific significance of the study by providing potential chemotaxonomic insights. If no clear markers are identified, the authors may discuss possible reasons or limitations in detecting species-specific compounds.

Response 8: Our analysis of the essential oil compositions revealed that certain compounds, such as monoterpenes, were present across all species, reflecting the genus's characteristic chemical profile. However, we did not identify unique chemical markers exclusive to any single species that could serve as distinguishing features.​ We have incorporated this discussion into the manuscript to provide a comprehensive understanding of the challenges in identifying chemical markers for species differentiation within the Heracleum genus.

Comment 9: In the Methods section, the duration of methanol extraction using the maceration technique has not been specified.

Response 9: Revision have been made.

Comment 10: In the solvent extraction section, it is unclear why only the root and aerial parts were extracted, while the fruit was not included. The authors should provide a rationale for this decision.

Response 10: In our study, we excluded fruits from the solvent extraction process due to an insufficient quantity of fruit samples available for both analyses. The limited fruit samples we had were prioritized for essential oil analysis, as this was a primary focus of our research.

Comment 11: In the antioxidant study, for both experiments, the researchers should provide a clear explanation of the negative control used in this study.

Response 11: Revision have been made.

Comment 12: In the study on antibiofilm and anti-quorum sensing activities, the authors have not provided details regarding the positive and negative controls used in the experiments. To ensure methodological transparency and facilitate result interpretation, the authors should specify the positive and negative controls in both the Methods and Results sections.

Response 12: As stated in the Materials and Methods section, Brain Heart Infusion (BHI) broth enriched with 2% sucrose was used as the negative control in our antibiofilm assays.

Luria Bertani (LB) broth was used as the negative control in our anti-QS assays. To avoid visual confusion in the figures, these controls were conducted on separate plates and were not included in the presented images. However, if desired, we can provide these images as supplementary material.

Comment 13: Several typographical and formatting errors have been identified, including incorrect scientific name formatting. For example, in line 297, the word Heracleum refers to a plant genus and should therefore be italicized. The authors are encouraged to carefully proofread the manuscript to ensure proper spelling, grammar, and adherence to scientific writing conventions

Response 13: We have carefully proofread the manuscript and corrected all typographical and formatting errors, including the italicization of scientific names such as Heracleum in line 297. We have ensured adherence to proper spelling, grammar, and scientific writing conventions throughout the revised manuscript.

Round 2

Reviewer 2 Report

Comments and Suggestions for Authors

Thank you for the authors for considering my comments. There is a significant improvement in the result presentation and discussion; however, I still have several comments which, in my opinion, were not fully clarified: 

  1. I'm still confused with the yield of methanolic extract (STable 1). For instance for H. crenatifolium aerial parts you have calculated 6.41 g/g [i.e. obtained extract was 6.41 g from 1 g of drug?]. Could you describe in materials and method parts the calculation method?
  2. Although SD values of biofilm inhibition would be expected on the diagrams, the raw data of ODs are welcome. Make sure to include in sequential numbering these tables and add in-text reference.
  3. In Materials and Methods section you must provide details about applied instruments (i.e. brand and producer of spectrophotometer).
  4. Check the Authors guideline of the journal and modify the in-text references of Tables and Figures moved to Supplementary material (https://www.mdpi.com/journal/plants/instructions#preparation, Section: Back matter).

Author Response

Responses to Reviewer 2 Comments

Thank you very much for taking the time to review this manuscript. Please find the detailed responses below and the corresponding revisions in track changes in the re-submitted files.

Comment 1: Thank you for the authors for considering my comments. There is a significant improvement in the result presentation and discussion; however, I still have several comments which, in my opinion, were not fully clarified:

I'm still confused with the yield of methanolic extract (STable 1). For instance for H. crenatifolium aerial parts you have calculated 6.41 g/g [i.e. obtained extract was 6.41 g from 1 g of drug?]. Could you describe in materials and method parts the calculation method?

Response 1: Thank you for your valuable feedback. We acknowledge the confusion regarding the unit representation in Table 1. The correct unit should be g/100 g, and we have now revised the table accordingly.

Additionally, while the Materials and Methods section already states that the extraction yield is calculated based on 100 g of dried plant material, the incorrect unit in the table may have led to misunderstandings. This has now been corrected to ensure consistency and clarity.

Comment 2: Although SD values of biofilm inhibition would be expected on the diagrams, the raw data of ODs are welcome. Make sure to include in sequential numbering these tables and add in-text reference.

Response 2: We have sequentially numbered the supplementary tables to ensure consistency. In-text references have been added to direct readers to the relevant supplementary material.

Comment 3: In Materials and Methods section you must provide details about applied instruments (i.e. brand and producer of spectrophotometer).

Response 3: We have now updated the Materials and Methods section to include the brand and model of the spectrophotometer used for total phenol and flavonoid content determinations: Shimadzu UV Spectrophotometer UV-1800.

Comment 4: Check the Authors guideline of the journal and modify the in-text references of Tables and Figures moved to Supplementary material (https://www.mdpi.com/journal/plants/instructions#preparation, Section: Back matter).

Response 4: We have carefully reviewed the Author Guidelines of the journal and have updated the in-text references for the Tables and Figures moved to the Supplementary Material in accordance with the journal's formatting requirements.

Reviewer 3 Report

Comments and Suggestions for Authors

After reviewing the revised manuscript, the author has responded to the suggestions and addressed the questions I raised very well. I believe this manuscript can be accepted for publication in Pharmaceuticals.

Author Response

Responses to Reviewer 3 Comments

Comment 1: After reviewing the revised manuscript, the author has responded to the suggestions and addressed the questions I raised very well. I believe this manuscript can be accepted for publication in Pharmaceuticals.

Response 1: Thank you for your positive feedback and for recognizing our efforts in addressing the suggestions and questions raised. We truly appreciate your time and valuable insights, which have helped improve the quality of our manuscript.
